# MicroRNA in Papillary Thyroid Carcinoma: A Systematic Review from 2018 to June 2020

**DOI:** 10.3390/cancers12113118

**Published:** 2020-10-25

**Authors:** Liviu Hitu, Katalin Gabora, Eduard-Alexandru Bonci, Andra Piciu, Adriana-Cezara Hitu, Paul-Andrei Ștefan, Doina Piciu

**Affiliations:** 1Doctoral School, Iuliu Hațieganu University of Medicine and Pharmacy, 400012 Cluj-Napoca, Romania; gabora.katalin@yahoo.com (K.G.); bonci.eduard@gmail.com (E.-A.B.); stefan_paul@ymail.com (P.-A.Ș.); doina.piciu@gmail.com (D.P.); 2Department of Medical Oncology, Iuliu Hatieganu University of Medicine and Pharmacy, 400012 Cluj-Napoca, Romania; piciuandra@gmail.com; 3Department of Dermatology, Iuliu Hatieganu University of Medicine and Pharmacy, 400012 Cluj-Napoca, Romania; cezara.lala@gmail.com; 4Radiology and Imaging Department, County Emergency Hospital, Cluj-Napoca, Clinicilor Street, Number 3-5, Cluj-Napoca, 400006 Cluj, Romania; 5Department of Endocrine Tumors and Nuclear Medicine, Institute of Oncology “Prof.dr.Ion Chiricut,ă” 400015 Cluj-Napoca, Romania

**Keywords:** microRNA, miRNA, papillary, thyroid, carcinoma, microcarcinoma

## Abstract

**Simple Summary:**

The most common form of endocrine cancer - papillary thyroid carcinoma, has an increasing incidence. Although this disease usually has an indolent behavior, there are cases when it can evolve more aggressively. It has been known for some time that it is possible to use microRNAs for the diagnosis, prognosis and even treatment monitoring of papillary thyroid cancer. The purpose of this study is to summarize the latest information provided by publications regarding the involvement of microRNAs in papillary thyroid cancer, underling the new clinical perspectives offered by these publications.

**Abstract:**

The involvement of micro-ribonucleic acid (microRNAs) in metabolic pathways such as regulation, signal transduction, cell maintenance, and differentiation make them possible biomarkers and therapeutic targets. The purpose of this review is to summarize the information published in the last two and a half years about the involvement of microRNAs in papillary thyroid carcinoma (PTC). Another goal is to understand the perspective offered by the new findings. Main microRNA features such as origin, regulation, targeted genes, and metabolic pathways will be presented in this paper. We interrogated the PubMed database using several keywords: “microRNA” + “thyroid” + “papillary” + “carcinoma”. After applying search filters and inclusion criteria, a selection of 137 articles published between January 2018–June 2020 was made. Data regarding microRNA, metabolic pathways, gene/protein, and study utility were selected and included in the table and later discussed regarding the matter at hand. We found that most microRNAs regularly expressed in the normal thyroid gland are downregulated in PTC, indicating an important tumor-suppressor action by those microRNAs. Moreover, we showed that one gene can be targeted by several microRNAs and have nominally described these interactions. We have revealed which microRNAs can target several genes at once.

## 1. Introduction

Given that in the last two and a half years alone, more than 200 articles have been published on the involvement of microRNAs in the pathology of PTC, microRNAs are a hot topic. Several questions arise from the analysis of these studies, for example, how relevant is the information of these studies to daily medical practice; Will this information ever be beneficial for the patients with an aggressive form of papillary carcinoma? Or we design studies on the treadmill pursuing only scientific interest? During the 1970s, Francis Crick asserted what he believed to be the central dogma of molecular biology. Genetic information traveled from deoxyribonucleic acid (DNA) to ribonucleic acid (RNA) through transcription, then from RNA to proteins via translation, meaning proteins were the functional end products of genes. However, after the whole human genome sequencing, it was understood that genes that encoded proteins accounted for less than 2% of the genome. Given the intricacy of cellular processes, genetic information is most likely passed by additional regulatory elements and not only by coding genes [1]. MicroRNAs (miRNAs) are a class of endogenous non-coding RNA molecules ranging from 18 to 22 nucleotides in length. MicroRNA’s constitute only 3% of the human genome but it is believed that they regulate more than half of the protein-coding genes. Mature microRNA can promote or inhibit messenger ribonucleic acid (mRNA) translation and degradation by targeting with precision complementary sequences in 3′UnTranslated Regions (3′UTR). They act both as post-transcriptional regulators of gene expression and as messengers or intercellular signaling [2]. MicroRNAs are involved in central biological processes, including development, organogenesis, tissue differentiation, cell cycles, and metabolism. Alterations in the expression of microRNA contribute to the pathogenesis of the majority of human malignancies (PTC) [3,4]. The most striking evidence that links microRNAs with thyroid cancer is their large alteration in expression in malignant cells compared to benign cells. MicroRNA expression is dysregulated in human cancer through various mechanisms. The most important are amplification or deletion of microRNA genes, abnormal transcriptional control of microRNAs, and epigenetic changes [5]. PTC is the most common thyroid malignancy [6], and it is defined as a malignant epithelial tumor with evidence of follicular differentiation and a series of specific nuclear features [7]. The incidence of PTC is increasing mainly due to improved diagnostic methods such as ultrasound (US) with targeted fine-needle aspiration biopsy (FNAB) [8]. This increase has been predominantly an increase in diagnosing papillary thyroid microcarcinoma (PTMC). PTMC is defined as measuring 1.0 cm or less in the greatest dimension of the tumor [9]. Cervical lymph nodes, lungs, and bones are the most common metastatic sites, brain, liver, and skin involvement is less common. Distant metastases are usually diagnosed through clinical symptoms or suspicious imaging/laboratory findings (abnormal uptake on the post-ablation whole-body scan (WBS). Another diagnostic method can be a positive finding on an F18-fluorodeoxyglucose (F18-FDG) positron emission tomography/computed tomography (PET/CT) evaluation or cross-sectional study prompted by elevated thyroglobulin levels in patients whose post-ablation WBS is negative [10]. Usually, PTC has an excellent prognosis [11]. Therefore, what are the special situations in daily practice that make us need these new potential biological markers for PTC diagnosis, prognosis, and therapeutic targets? The purpose of this review is to summarize the information published in the last two and a half years about the involvement of microRNAs in PTC. It is also to understand the perspective offered by the new findings.

## 2. Materials and Methods

A literature analysis was performed in MEDLINE using PubMed for studies published from 2018 to June 2020. The following keywords were used: ”microRNA”+ ”papillary”+ “thyroid” + “carcinoma”, which resulted in 466 articles in English. All related abstracts were reviewed and relevant studies that were published in English were selected. We only included papers that had full text available and described the exact method and results regarding microRNA’s signatures in PTC epigenetic mechanism. A selection of 137 eligible articles was the result of our search (Figure 1).

## 3. Results

Data on microRNAs, the sample source, the regulatory mode of microRNAs, the target genes/proteins of microRNAs, and their effect on PTC cells from the 137 studies were selected and presented in Table 1 [12,13,14,15,16,17,18,19,20,21,22,23,24,25,26,27,28,29,30,31,32,33,34,35,36,37,38,39,40,41,42,43,44,45,46,47,48,49,50,51,52,53,54,55,56,57,58,59,60,61,62,63,64,65,66,67,68,69,70,71,72,73,74,75,76,77,78,79,80,81,82,83,84,85,86,87,88,89,90,91,92,93,94,95,96,97,98,99,100,101,102,103,104,105,106,107,108,109,110,111,112,113,114,115,116,117,118,119,120,121,122,123,124,125,126,127,128,129,130,131,132,133,134,135,136,137,138,139,140,141,142,143,144,145,146,147,148].

### 3.1. Up- and Downregulated microRNAs in Papillary Thyroid Cancer

Out of 139 microRNAs, 106 are downregulated and 33 are upregulated (Table 1). This means that more than a quarter of the described microRNAs have an oncogenic role (oncomiR’s) and the rest of them have a tumor-suppressive role. The dysregulation of microRNA is an important event during the development of papillary thyroid carcinoma. Overexpression of certain microRNA can result in the tumor suppressor genes repression. Down-regulation of specific microRNA can lead to increased expression of oncogenes. Overexpression and downregulation induce malignant effects on cell cycle progression, proliferation, migration, and apoptosis, leading to tumor growth and progression in PTC and other types of malignancies [1].

### 3.2. One Gene Can Be Targeted by Several microRNAs

Analyzing, individually, in each study, the interaction between microRNAs and the genes targeted by them, we noticed that the same gene can be targeted by different microRNAs. For example, HMGB1 has been reported to play an important role in promoting both cell survival and death by regulating multiple signaling pathways, including proliferation, autophagy, inflammation, invasion, and metastasis. The study by Ding. C et al. [127] indicates that microRNA-let-7e downregulates HMGB1 expression by directly targeting the HMGB1 3′-UTR, downregulated HMGB1 inhibits PTC cell proliferation and metastasis [127]. MicroRNA-1179 interacted with the 3′ UTR of HMGB1 and suppressed HMGB1 expression at the post-transcriptional level and indicates that the microRNA-1179/MHGB1 pathway plays a tumor suppressor role in PTC [97]. The same gene-HMGB1 is involved in ANRIL/HMGB1/ microRNA-320a pathway. Propofol-mediated ANRIL downregulation competed with HMGB1 to bind microRNA-320a, thus inhibiting PTC cell malignant behaviors [42].

A study by Chen et al. [18] has shown that enforced expression of microRNA-202-3p inhibited WNT signaling by downregulating β-catenin expression in PTC. Again, the same gene is regulated by microRNA-3619-3p to promote cell migration and invasion in PTC [102]. WNT1 has been shown to promote cancer progression because it triggers cell proliferation and metastasis, microRNA-329 inhibits papillary thyroid cancer progression via direct targeting WNT1 [90]. WNT5a, an important signaling molecule in the non-canonical Wnt family, has been involved in nearly all parts of the non-canonical Wnt pathway. The invasion and metastasis of PTC cells were inhibited by microRNA-26a- 5p via Wnt5a [66].

B-cell lymphoma-2 (Bcl-2), an oncogene expressed in most thyroid carcinomas, is also found to be a target of several different microRNAs. MicroRNA-21-5p suppressed Bcl-2 expression [138], silencing LINC00313 led to down-regulation of anti-apoptotic Bcl-2 proteins [136]. Overexpression of miR microARN-203 may serve a role in PTC tumor cells by downregulating Bcl-2 expression [91].

One more targeted gene by multiple microRNAs in PTC is AKT, the human homolog of the viral oncogene v-Akt is related to protein kinases A (PKA) and C (PKC) in humans. The pathway that involves AKT inactivates several proapoptotic factors, AKT also activates transcription factors which promote anti-apoptotic genes. Overexpression of microRNA-15a inhibited the activation of the AKT pathway, which inhibited cell proliferation and promoted the process of apoptosis [39]. Upregulated microRNA-203 suppresses epithelial-mesenchymal transition (EMT), invasion, proliferation, and migration as well as induces apoptosis of PTC cells via downregulated AKT3 [101]. lncRNA n384546 could regulate the expression of AKT3 by sponging microRNA-145-5p [119]. lncRNA HOTTIP modulated Akt1 expression by regulating microRNA-637 in PTC cell lines [103].

Another example is the Sphingosine kinase (SPHK), an enzyme, catalyzing the formation of the prosurvival second messenger sphingosine-1-phosphate (S1P) from the pro-apoptotic lipid sphingosine. High SPHK expression is correlated with a significant decrease in survival rate in patients with several forms of cancer, including PTC. LncRNA LINC00460 promoted PTC progression via modulating SphK2 through sponging microRNA-613 in PTC [26]. lncRNA LINC00520 accelerates the progression of papillary thyroid carcinoma by serving as a competing endogenous RNA of microRNA-577 to increase SphK2 expression [72]. MicroRNA-128 targets SPHK1 to induce apoptosis and reduce cell proliferation, migration in thyroid cancer cell lines, and inhibits tumor growth [14].

PTEN (phosphatase with tensin homology), an upstream negative regulatory molecule of the PI3K/AKT pathway, is the direct target gene of microRNA-106 [116] and microRNA-21 [134]. MicroRNA-625-3p [24] and microRNA-564 [67] directly target the same gene, AEG-1 (astrocyte elevated gene 1), an important regulator of PTC genesis and development. Yes-associated protein 1 (YAP1) was identified as a target gene of microRNA -205 [44] and microRNA-200a-3p [89]. Both microRNA-361-5p [46] and microRNA-26a [92] target ROCK1 (Rho-associated coiled-coil kinase 1), which was closely associated with poor PTC prognosis [46]. Zinc Finger E-Box Binding Homeobox 1 (ZEB1) a gene that plays vital roles in the metastasis of cancer, is inhibited by microRNA-451a [80] and a direct target of microRNA-429 [31].

From the same families of genes discovered to be the target of several microRNAs we mention CXCL-12/16 [16,86], CCNG-1/2 [17,100], IRS-1/2 [19,107], FGF-2/FR [28,112], CCDC-6/67 [52,128], FOX-O1/E1/M1 [27,55,76], ITGA-3/6 [55,61], SL-1A5/25A1 [56,118], MAP-2K4/K1/4K3 [71,78,120], and SOX 11/12/2 [84,95,117].

### 3.3. One MicroRNA Can Target Several mRNAs/Genes

One microRNA does not target only one but several mRNA/genes, as it was stated before. In our study, we found several microRNAs with multiple genes targeted. For instance, microRNA-146b-5p, downregulated CCDC6 expression by binding to its 3′-UTR in the study by Jia et al. [128] and promoted the expression of MALAT1 by negatively regulating DNMT3A in the study by Peng et al. [63]. The same microRNA-146b, but with the 3p strand located in the reverse position compared to the 5p strand which is present in the forward (5′-3′) position, meaning microRNA-146b-3p, is targeting directly NF2 [132]. From the same family, microRNA-146a targets GABPA [58], and microRNA-146 targets KIT [114]. MicroRNA-199a-5p inhibited cell migration, invasion, and EMT by targeting SNAI in PTC [60] but also attenuated cell proliferation, induced apoptosis, and arrested cells in the G0/G1 phase through regulating the expression of SLC1A5 [118]. From the same family, microRNA-199b-5p suppressed PTC cell aggressiveness by targeting STON2 [65]. MicroRNA-101 suppresses the proliferation, apoptosis resistance, invasion, and migration of PTC cells by directly targeting CXCL12 [16]. MicroRNA-101-3p deficiency enhanced the expression level of FN1, which therefore promoted the RAI (radioactive iodine)-resistance of PTC [49]. Another example is microRNA-150 which serves a key function in suppressing the malignant growth and aggressive behavior of PTC cells through the downregulation of MUC4 [22]. Overexpression of microRNA-150-5p regulated cell proliferation, metastasis, and apoptosis by regulating BRAFV600E [94]. Both the IGF1R [139] and E2F7 [32] genes are targeted directly by the microRNA-30a. MicroRNA-203 inhibits proliferation and motility, and induces apoptosis of PTC cells via regulation of the expression of Bcl-2 [91], and suppresses EMT, invasion, proliferation, and migration of PTC cells via downregulated AKT3 [101]. Five more genes are found to be the target of the same microRNA in four different studies. VHL [104], Bcl-2 [138], PTEN [134], TGFBI [122], and COL4A1 [122] are all targeted genes by microRNA-21. There are still more examples of the same microRNA’s but with the 3p strand located in the reverse position compared to 5p strand which is present in the forward (5′-3′) position, that target different genes: [52,108], [119,129], [29,140], [37,73], [39,131], [45,135], [48,120], [50,68], [66,92], and [57,117].

## 4. Discussion

Each microRNA can regulate hundreds of messenger RNAs (mRNAs), while various microRNA can control the same mRNA. Additionally, many microRNAs regulate and are regulated by other species of non-coding RNAs, such as circular RNAs (circRNAs) and long non-coding RNAs (lncRNAs). For this reason, it is extremely difficult to predict, study, and analyze the precise role of a single microRNA involved in human cancer, considering the complexity of its connections. Focusing on a single microRNA molecule represents a limited approach. Additional information could come from network analysis, which has become a common tool in the biological field to better understand molecular interactions [1].

Most studies assess the level of expression of the microRNA in question, show the actions of its overexpression/silencing on cell lines, find the gene targeted by the microRNA, and how the metabolic pathway microRNA / target gene works. Although complex information is presented, at the end of the discussion chapter we find the same dry phrase “microRNA-X could be a potential therapeutic/diagnostic/prognostic target for PTC treatment”. Despite this, there are several articles with a different study design that offer something more than “could be”. One of them is the study of Zhao. L et al. [134] which finds Matrine, a traditional Chinese medicine, as an alternative drug for PTC treatment. Treatment with matrine at the concentrations of 1, 2, 5, 10, and 20 mg/ml inhibited TPC 1 cell proliferation by up to 95.8% (for 20 mg/ml matrine). Matrine induced apoptosis and G1 cell cycle arrest through downregulating microRNA-21 to affect the PTEN/Akt signaling in TPC 1 human thyroid cancer cells. Liu. F et al. [121] discovered that microRNA-206 contributed to euthyrox resistance in PTC cells through blockage p38 and JNK signaling pathway by targeting MAP4K3. Another study by Liu et al. [49], found the promoting gene and the signaling pathway regulating RAI-resistance in PTC. The results attested that NEAT1 was upregulated in RAI-resistant PTC accompanied microRNA-101-3p inhibition, FN1 overexpression, and PI3K/AKT signaling pathway abnormal activation. Fang. T et al. [23] discovered that Shenmai injection (SMI), a traditional Chinese formula mainly made up of Red Ginseng and Radix Ophiopogonis. SMI inhibits the differentiation of CD4 + T cells into Treg cells via the microRNA-103/GPER1 axis, which improves the immunological function of PTC patients with postoperative 131-Iodine ablation. Although few, these studies differ in the classical approach to the use of microRNAs in papillary thyroid carcinomas and should be recognized as at least promising.

Even if in the world of publications microRNA is a hot topic, when we talk about PTC, most international guidelines regarding thyroid cancer management, do not even mention microRNA. Here we refer to the NCCN 2018 [149], ETA 2019 [7], and ESMO 2019 [11] guidelines. The exception is the ATA 2015 guideline, which, although published several years before the above-listed guidelines, mentions microRNA as an additional diagnostic molecular marker strategy under development. microRNA markers have shown initial diagnostic utility in FNA samples with indeterminate cytological diagnoses, but they have not been thoroughly validated. It is also mentioned about microRNA, also in this guide, in the chapter “Directions for future research”, as possible progress in identifying markers of thyroid cancer. To result in a significantly improved accuracy of cancer detection in thyroid nodules as compared to the currently available clinical tests [8].

Hence, which are the most challenging parts in PTC management where we could use microRNA? After the clinical and ultrasound evaluation of a thyroid nodule, if malignancy criteria are present, a fine needle biopsy is performed for cytological examination. Some results of the cytological examination can be inconclusive. In such cases, there is a need to assess molecular markers to make a presurgical differentiation of benign and malign lesions. MicroRNAs are one of the novel classes of molecular markers that are being used to improve the diagnosis of thyroid cancer. Several studies have shown that a microRNA-based signature in FNABs can be used to discriminate benign from malignant thyroid nodules. MicroRNA profiling of thyroid cancers can also provide prognostic information useful for defining optimal management strategies. Expression levels of certain microRNA in thyroid tumor tissues are associated with clinicopathological characteristics, such as tumor size, multifocality, capsular invasion, extrathyroidal extension, and both lymph node and distant metastases [150]. Treatment options have been proposed and implemented based on the results obtained from research conducted on epigenetic alterations. Therefore, the development of new therapeutic strategies based on targeting epigenetic changes to restore the expression of tumor suppressor microRNAs or to blunt overexpressed oncogenic microRNAs may provide a new landscape for the treatment of aggressive PTC [151].

Although PTMC generally has an excellent prognosis, the long-term rate of recurrence of PTMC has been reported to be as high as 10% [9]. Currently, there are no reliable clinical features including molecular markers, that can differentiate PTMC in patients who develop progressive disease from indolent PTMC. Even so, searching the PubMed database, regarding microRNA signatures in PTMC, there is only one study by Zhang et al. which combines serum microRNA with ultrasound profile as predictive biomarkers of diagnosis and prognosis for PTMC. In this study, microRNAs were found to be significantly associated with a poor prognosis of patients with PTMC and could be used as prognostic molecular markers or patients with PTMC before and after surgery. These results suggest that circulating microRNAs may be useful as non-invasive molecular biomarkers of diagnosis and prognosis for PTMC [9], selecting those cases that need aggressive therapies, despite the histology of PTMC. Given the need for more studies in this field, this topic could be a research idea for the future, in the era of personalized medicine.

## 5. Conclusions

Research regarding microRNAs in PTC is undergoing a tremendous shift, suggesting rapid maturation of this field. In this review, we tried to represent as briefly as possible the interactions of microRNAs with target proteins. We also showed microRNAs regulation mode and its effect on PTC cells. Our results showed that a gene can target multiple microRNAs simultaneously, and vice versa. All this information can be used to find the most effective therapeutic targets/biomarkers in PTC. For future research, we indicated a possible niche, namely microRNA signatures in PTMC.

## Figures and Tables

**Figure 1 cancers-12-03118-f001:**
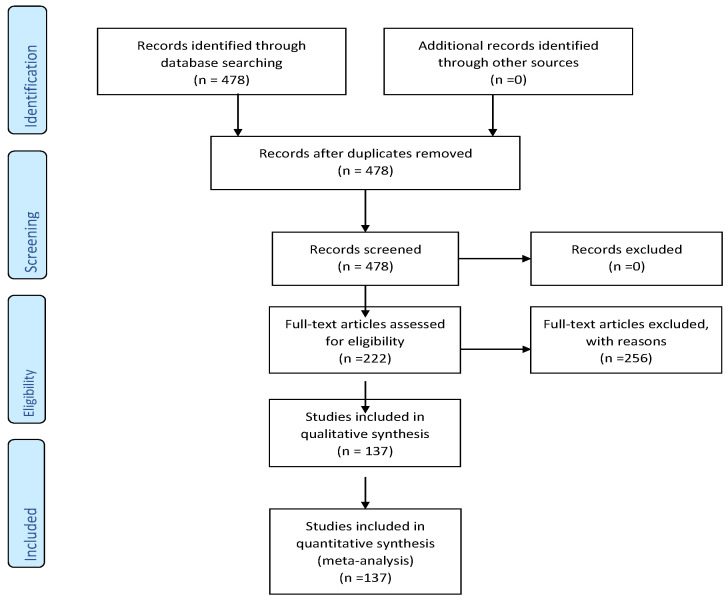
Study selection summary.

**Table 1 cancers-12-03118-t001:** Targets of epigenetic alterations in papillary thyroid carcinoma (PTC).

miRNA	Sample Sources	Up/Down-Regulated	Targeted Gene	miRNA Effect on PTC Cells	Sponging/Sequestering	Potential Utility	References
miR-520a-3p	tissue	Down	JAK1	prevented cell proliferation, migration, invasion, promoted apoptosis		TT	Bi. CL et al. [12]
miR-139-5p	tissue	Down	LMO4	suppressed cell proliferation, migration, and tumor growth	circBACH2	Pr, TT	Cai. X et al. [13]
miR-128	tissue	Down	SPHK1	elevated apoptosis increased G0/G1 arrest, reduced Cyclin D1/ CDK4 expressions	-	TT	Cao. XZ et al. [14]
miR-2861	tissue	Up	DGCR5	promoted cell proliferation and invasion	lncRNA DGCR5	Pg	Chen. F et al. [15]
miR-101	tissue	Down	CXCL12	repressed cell proliferation, migration, and invasion. Enhanced apoptosis	-	Pg, TT	Chen. F et al. [16]
miR-135b-5p	tissue	Up	CCNG2	modulated tumor cells proliferation, apoptosis, migration	lncRNA GAS8-AS1	TT	Chen. N et al. [17]
miR-202-3p	tissue	Down	WNT	suppressed the expression of β-catenin, cell migration, and invasion	-	TT	Chen. J et al. [18]
miR-1271	tissue	Up	IRS1	suppressed migration, invasion, and proliferation of PTC cells	-	TT	Chen. Y et al. [19]
miR-153-3p	tissue	Down	ZNRF2	regulated cell proliferation, migration, and invasion	lncRNA TTN-AS1	TT	Cui. Z et al. [20]
miR-548c-3p	tissue	Down	HIF1α	reducted N-cadherin and vimentin expression	-	Pg, TT	Du. Y et al. [21]
miR-150	tissue	Down	MUC4	suppressed PTC cell proliferation and metastasis	-	Dg, TT	Fa. Z et al. [22]
Ex-miR-103	blood	Up	GPER1	influenced the differentiation of CD4 + T cell into Treg cells	-	TT	Fang. T et al. [23]
miR-625-3p	tissue	Up	AEG-1	promoted the proliferation, migration, and invasion of thyroid cancer cells	-	TT	Fang. L et al. [24]
miR-141-3p	tissue	Down	YY-1	inhibited cell growth, induced apoptosis, and suppressed invasion	-	Pg, TT	Fang. M et al. [25]
miR-613	tissue	Down	SphK2	inhibited cell proliferation, migration, and invasion	lncRNA LINC00460	TT	Feng. L et al. [26]
miR-93-3p-660	tissue	Up	FOXO1	inhibited glycolysis that attenuated glucose uptake and lactate production	lncRNA ASMTL-AS1	TT	Feng. Z et al. [27]
miR-1266	tissue	Down	FGFR2	inhibited PTC cell proliferation, migration, and invasion	-	Dg, TT	Fu. YT et al. [28]
miR-129	tissue	Up	MAL2	suppressed growth and invasion of PTC cells	-	Pv, TT	Gao. X et al. [29]
miR-791	tissue	Down	-	inhibited proliferation of PTC cells via blocking the G1 phase	-	Pg, TT	Gao. XB et al. [30]
miR-429	tissue	Down	ZEB1	inhibited cell proliferation, migration, and invasion	lncRNA SNHG22	Dg, TT	Gao. H et al. [31]
miR-30a	tissue	Down	E2F7	inhibited the proliferation, migration, and invasion of PTC cells	-	TT	Guo. H et al. [32]
miR-9-5p	tissue	Down	BRAF	suppressed the viability of PTC cells by inducing apoptosis	-	TT	Guo. F et al. [33]
miR-215	tissue	Down	ARFGEF1	inhibited proliferation and metastasis	-	Pg	Han. J et al. [34]
Ex-miR-199	blood	Down	DLG1-AS1	suppressed proliferation of PTC cells	-	Dg	He. T et al. [35]
miR-1252	tissue	Up	FSTL1	inhibited viability, proliferation, and stimulated apoptosis in PTC cells.	hsa_circ_ 0011290	TT	Hu. Z et al. [36]
miR-486-5p	tissue	Down	KIAA1199	inhibited cell growth of PTC	-	TT	Jiao. X et al. [37]
miR-885-5p	tissue	Down	RAC1	suppressed PTC cell proliferation	hsa_circ_ 0004458	TT	Jin. X et al. [38]
miR-15a	tissue	Down	RET/AKT	inhibited PTC cell proliferation and invasion and enhanced the apoptosis	-	TT	Jin. J et al. [39]
miR-381-3p	tissue	Down	LRP6	inhibited the proliferation and metastasis of PTC cells	-	TT	Kong. W et al. [40]
miR-485-5p	tissue	Down	Raf1	inhibited PTC cell proliferation, migration and invasion	LncRNA LINC00460	TT	Li. G et al. [41]
miR-320a	cell culture	Up	HMGB1	inhibited cell proliferation, migration and invasion rates	lncRNA ANRIL	Pv, TT	Li. M et al. [42]
miR-369-3p	cell culture	Down	TSPAN13	suppressed cell proliferation, colony formation, and induced apoptosis in PTC	-	Dg, Pv, TT	Li. P et al. [43]
miR-205	tissue	Down	YAP1	inhibited proliferation and invasion of thyroid cancer cells	-	TT	Li. D et al. [44]
miR-204-3p	tissue	Down	CDC23	suppressed PTC proliferation, migration and invasion	lncRNA LINC00514	TT	Li. X et al. [45]
miR-361-5p	tissue	Down	ROCK1	Inhibited proliferation, migration, and invasion	-	TT	Li. R et al. [46]
miR-4500	cell culture	Down	PLXNC1	inhibited cell viability, colony formation, and cell apoptosis	-	Pg, TT	Li. R et al. [47]
miR-211-5p	tissue	Down	SPARC	suppressed proliferation, migration, and invasion of thyroid tumor cells	lncRNA MCM3AP-AS1	TT	Liang. M et al. [48]
miR-101	tissue	Down	FN1	promoted the RAI-resistance in PTC	lncRNA-NEAT1	TT	Liu. C et al. [49]
miR-214	tissue	Down	PSMD10	suppressed proliferation, and induced cell apoptosis and cell cycle arrest in PTC cells	-	TT	Liu. F et al. [50]
miR-744	tissue	Down	NOB1	attenuated the proliferation and invasion of PTC cells	-	TT	Liu. H et al. [51]
miR-96-5p	tissue	Up	CCDC67	accelerated the proliferation and metastasis of PTC cells	-	Dg, TT	Liu. ZM et al. [52]
miR-4728	cell culture	Down	SOS1	repressed the PTC cell proliferation through MAPK	-	Dg, TT	Liu. Z et al. [53]
miR-431	tissue	Down	Gli1	inhibited cell migration and invasion of PTC	-	TT	Liu. Y et al. [54]
miR-524-5p	tissue	Down	FOXE1, ITGA3	suppressed PTC progression by regulating tumor cell proliferation, migration, and invasion	-	TT	Liu. H et al. [55]
miR-331-3p	tissue	Down	SLC25A1	inhibited PTC cells proliferation, migration and invasion	lncRNA-BRM	TT	Liu. S et al. [56]
miR-335-5p	tissue	Down	ICAM1	reduced the proliferation, migration, invasion of PTC	-	TT	Luo. L et al. [57]
miR-146a	tissue	-	GABPA	suppressed proliferation, migration, and invading capabilities of PTC cells		TT	Long. M et al. [58]
miR-29a-3p	tissue	Down	OTUB2	suppressed growth, proliferation, invasion in PTC cells.	-	Pg, TT	Ma. Y et al. [59]
miR-199a-5p	tissue	Down	SNAI1	reduced migration and invasion of PTC cells	-	TT	Ma. S et al. [60]
miR-363-3p	tissue	Down	ITGA6	suppressed anoikis resistance in PTC cells	-	TT	Pan. Y et al. [61]
miR-1231, miR-1304	tissue	Down	-	inhibited proliferation and invasion of PTC cells	circ_ 0025033	TT	Pan. Y et al. [62]
miR-146b-5p	tissue	Up	DNMT3A	accelerated extra-glandular invasion and metastasis of PTC cells	lncRNA-MALAT1	Dg, TT	Peng. Y et al. [63]
miR-448	tissue	Down	KDM5B	inhibited PTC cell progression and tumor growth via TGIF1	-	TT	Pu. Y et al. [64]
miR-199b-5p	tissue	Down	STON2	inhibited PTC cell growth and metastasis	-	TT	Ren. L et al. [65]
miR-26a-5p	tissue	Down	Wnt5a	inhibited proliferation, colony formation, invasion, and migration of PTC cells.	-	TT	Shi. D et al. [66]
miR-564	tissue	Down	AEG-1	inhibited cell proliferation, migration, and invasion and induced cell apoptosis	-	TT	Song. Z et al. [67]
miR-214-3p	tissue	Down	PSMD10	impaired PTC cell proliferation and metastasis	lncRNA-SNHG3	TT	Sui. G et al. [68]
miR-144	tissue	Down	WWTR1	inhibited of PTC cell proliferation	-	Pg, TT	Sun. W et al. [69]
miR-106b-5p	tissue	Down	ATAD2	induced apoptosis and suppressed invasion of PTC cells	lncRNA-NEAT1_2	TT	Sun. W et al. [70]
miR-124-3p	cell culture	Down	MAP2K4	inhibited the proliferation, induced apoptosis and cell cycle arrest in PTC cells	-	TT	Sun. Y et al. [71]
miR-577	tissue	Down	Sphk2	inhibited PTC cell proliferation, migration, and invasion	lncRNA-LINC00520	Dg, TT	Sun. Y et al. [72]
miR-486	tissue	Down	TENM1	inhibited proliferation, invasion, and migration of PTC cell	-	TT	Sun. YH et al. [73]
miR-497	tissue	Down	BDNF	suppressed PTC cell proliferation, migration, and invasion	lncRNA-LINC00152	TT	Sun. Z et al. [74]
miR-22	tissue	Up	-	promoted PTC cell metastasis and BRAFV600E mutation	-	Dg, Pg	Wang. D et al. [75]
miR-599	tissue	Down	Hey2	diminished PTC cell proliferation, migration, invasion, while stimulating apoptosis	-	Pr, TT	Wang. DP et al. [76]
miR-3619-5p	cell culture	Down	FOXM1	regulated proliferation and apoptosis in PTC	lncRNA-Linc01410	TT	Wang. G et al. [77]
miR-675	tissue	Down	MAPK1	suppressed PTC cell proliferation, migration, and invasion	lncRNA-RMRP	TT	Wang. J et al. [78]
miR-1258	cell culture	Down	TMPRSS4	inhibited cell viability, migration, and invasion	-	Dg, TT	Wang. L et al. [79]
miR-451a	tissue	Down	ZEB1	suppressed proliferation, mobility, and invasion of PTC cell	-	TT	Wang. Q et al. [80]
miR-622	tissue	Down	VEGFA	inhibited PTC cell proliferation, migration, and invasion	-	TT	Wang. R et al. [81]
miR-212	tissue	Down	MIAT	inhibited PTC cell proliferation, migration, and invasion.	(possible) lncRNA-MIAT	TT	Wang. R et al. [82]
miR-718	tissue	Down	PDPK1	inhibited PTC cell growth, reduced cell invasion, repressed glucose metabolism	-	Dg, TT	Wang. X et al. [83]
miR-31	tissue	Down	SOX11	repressed PTC cell proliferation, invasion, and migration	-	TT	Wang. Y et al. [84]
miR-384	tissue	Down	PRKACB	suppressed PTC cell proliferation and migration	-	TT	Wang. Y et al. [85]
miR-873	tissue	Down	CXCL16	inhibited proliferation, migration, and invasion of the PTC cells	-	TT	Wang. Z et al. [86]
miR-143-3p	tissue	Down	MSI2	induced apoptosis, suppresses invasion and migration of PTC	-	TT	Wang. ZL et al. [87]
miR-1261	tissue	Down	C8orf4	inhibited PTC cell proliferation, migration, and invasion	circZFR	TT	Wei. H et al. [88]
miR-200a-3p	tissue	Down	YAP1	inhibited PTC cell proliferation and promoted apoptosis	lncRNA-SNHG15	TT	Wu. DM et al. [89]
miR-329	tissue	Down	WNT1	decreased PTC cell proliferation, colony formation, suppressed growth	-	TT	Wu. L et al. [90]
miR-203	tissue	Down	Bcl-2	inhibited cell proliferation, induced apoptosis, and suppressed the motility of PTC cells	-	TT	Wu. X et al. [91]
miR-26a	tissue	Down	ROCK1	repressed PTC cell viability, invasion, and metastasis	-	Dg, TT	Wu. YC et al. [92]
miR-222	tissue	Up	-	correlated with capsular invasion, vascular invasion, tumor size and metastasis	-	Pg	Xiang. D et al. [93]
miR-150-5p	cell culture	Up	BRAF(V600E)	promoted PTC cell proliferation, suppressed apoptosis	-	TT	Yan. R et al. [94]
miR-423-5p	cell culture	Down	SOX12	suppressed PTC cell proliferation and invasion	lncRNA-NR2F1-AS1	TT	Yang. C et al. [95]
miR-182	tissue	Up	CHL1 *	correlated with extrathyroidal invasion, cervical lymph node metastasis, and TNM	-	Pg	Yao. XG et al. [96]
miR-1179	tissue	-	HMGB1	-	circFOXM1	TT	Ye. M et al. [97]
miR-1270	cell culture	Up	SCAI	promoted PTC cell proliferation, migration	-	TT	Yi. T et al. [98]
miR-761	tissue	Down	TRIM29	inhibited cell proliferation, and cell cycle progression in PTC	lncRNA- HOXA11-AS	TT	Yin. X et al. [99]
miR-23a-	tissue	Down	CCNG1	decreased proliferation, induced cell cycle arrest, and promoted PTC cell apoptosis	-	Dg, TT	Yin. JJ et al. [100]
miR-203	tissue	Down	AKT3	suppressed cell migration and invasion in the PTC cells and promoted cell apoptosis	-	TT	You. A et al. [101]
miR-3619-3p	tissue	Up	Wnt	promoted PTC cell migration and invasion	-	TT	Yu. S et al. [102]
miR-637	tissue	Down	Akt1	inhibit inhibited PTC cell proliferation, invasion, and migration	lncRNA HOTTIP	Dg, TT	Yuan. Q et al. [103]
miR-21	tissue	Up	VHL	promoted PTC cell proliferation and invasion	-	TT	Zang. C et al. [104]
miR-224-5p	tissue	Up	EGR2	promoted PTC cell migration, invasion	-	Dg, TT	Zang. CS et al. [105]
miR-509	tissue	Down	PAX9	inhibited cell proliferation and invasion in papillary thyroid carcinoma	-	TT	Zhang. S et al. [106]
miR-766	tissue	Down	IRS2	inhibited proliferation of PTC cells	-	TT	Zhao. J et al. [107]
miR-96-3p	tissue	Up	SDHB	increased the invasion and migration of PTC cells		TT	Zhao. X et al. [108]
miR-138-5p	tissue	Down	LRRK2	inhibited PTC cell proliferation, apoptosis	lncRNA RP11-476D10.1	TT	Zhao. Y et al. [109]
miR-409-3p	tissue	Down	CCND2	negatively regulated PTC cell proliferation and cell cycle progression	-	Pr, TT	Zhao. Z et al. [110]
miR-200b/c	tissue	Down	Rap1b	inhibited PTC cell invasion, migration and growth	-	TT	Zhou. B et al. [111]
miR-188-5p	tissue	Down	FGF-5 *	suppressed PTC cells growth	-	TT	Zhou. P et al. [112]
miR-506	tissue	Down	IL17RD	inhibited the proliferation, invasion, and migration capacities of PTC cells	-	Pg, TT	Zhu. J et al. [113]
miR-146	tissue	Up *	KIT	promoted PTC cell proliferation and invasion *	lncRNA CTC	TT	Liao. B et al. [114]
miR-1178	tissue	Up	TLR4	promoted cell proliferation and suppressed cell apoptosis	circ_FNDC3B	TT	Wu. G et al. [115]
miR-106a	tissue	Up	PTEN/ SFR4	enhanced PTC cell proliferative, migratory and invasive abilities	lncRNA-HULC	TT	Yang. Z et al. [116]
miR-335	tissue	Down	SOX2	suppressed the proliferation, migration, and invasion of PTC cells	lncRNA-LINC01510	TT	Li. Q et al. [117]
miR-199a-5p	tissue	Down	SLC1A5	attenuated proliferation, induced apoptosis, and arrested cells in the G0/G1 phase	ABHD11-AS1	Dg, TT	Zhuang. X et al. [118]
miR-145-5p	tissue	Down	AKT3	inhibited proliferation, migration and invasion	lncRNA-n384546	Dg, TT	Feng. J et al. [119]
miR-211	tissue	Up	RECK	promoted tumor growth and increased tumor volume in PTC cells	-	Dg, Pg	Wei. ZL et al. [120]
miR-206	tissue	Down	MAP4K3	enhanced Euthyrox sensitivity in Euthyrox-resistant PTC cells	-	TT	Liu. F et al. [121]
miR-21-5p	cell culture	Up	TGFBI, COL4A1	secreted by hypoxic PTC cells is a potent pro-angiogenic factor	-	Dg, TT	Wu. F et al. [122]
Ex-miR-423-5p	blood	Up	-	promoted PTC cell migration and invasion	-	Dg, TT	Ye. W et al. [123]
miR-422a	tissue	Down	SP1	suppressed PTC cells proliferation and metastasis	lncRNA-LINC00313	TT	Yan. D et al. [124]
miR-1301-3p	tissue	Down	STAT3	inhibited PTC cell proliferation, cell apoptosis -accelerated	lncRNA-ABDH11-AS1	TT	Wen. J et al. [125]
miR-let-7a	tissue	Down	c-Myc	suppressed PTC cell proliferation, migration, and invasion	-	TT	Huang. J et al. [126]
miR-let-7e	tissue	Down	HMGB1	inhibited of PTC cell growth and metastasis	-	TT	Ding. C et al. [127]
miR-146b-5p	tissue	Up	CCDC6	promoted proliferation, migration, invasion, and cell cycle progression of PTC cells	-	Dg, TT	Jia. M et al. [128]
miR-145	tissue	Down	ZEB2	inhibited the migration, invasion, and tumorigenesis of PTC cells	circ_NUP214	TT	Li. X et al. [129]
miR-520c-3p	tissue	Down	S100A4	inhibited PTC cells proliferation	lncRNA-HOXA-AS2	TT	Xia. F et al. [130]
miR-15a-5p	tissue	Down	-	inhibited PTC cells growth	lncRNA-HOXA-AS2	TT	Jiang. L et al. [131]
miR-146b-3p	tissue	Up	NF2	increased PTC cell migration and invasion	-	TT	Yu. C et al. [132]
miR-22a-3p	tissue	Up	CBL	promoted PTC cell proliferation and invasion	circ_ITCH	TT	Wang. M et al. [133]
miR-21	cell culture	-	PTEN	matrine- induced apoptosis and G1 cell cycle arrest	-	TT	Zhao. L et al. [134]
miR-204	tissue	Down	BRD4	inhibited of PTC cell proliferation	lncRNA-UCA1	TT	Li. D et al. [135]
miR-4429	tissue	Down	Bcl-2	suppressed PTC cell proliferation, promoted apoptosis, and induced cell cycle arrest in G2/M phase	lncRNA-LINC00313	TT	Wu. JW et al. [136]
miR-222	tissue	Up *	PPP2R2A	promoted PTC cell migration and invasion	-	Dg, TT	Huang. Y et al. [137]
miR-21-5p	tissue	Down	Bcl-2	inhibited TPC cell proliferation and invasion	lncRNA-BISPR	Dg, TT	Zhang. H et al. [138]
miR-30a	tissue	Down	IGF1R	inhibited PTC cell proliferation, cycle progression, invasion, migration	lncRNA-PVT1	TT	Feng. K et al. [139]
miR-129-5p	tissue	Down	KLK7	inhibited proliferation, cell survival, invasion, and migration	lncRNA-NEAT1	TT	Zhang. H et al. [140]
Ex-miR-146b-5p, Ex-miR-222-5p	blood	Down	-	enhanced the migration and invasion activity of PTC cells	-	Pg	Jiang. K et al. [141]
miR-539	tissue	Down	SLPI	inhibited PTC cell EMT and tumor growth	-	TT	Xu. CB et al. [142]
miR-24-3p	tissue	Up	p27kip1	regulated PTC cell proliferation, apoptosis migration and invasion	ncRNA-MIR22HG	TT	Chen. ZB et al. [143]
miR-26a	cell culture	Down	ARPP19	promoted proliferation of PTC cells	-	TT	Gong. Y et al. [144]
Ex-miR-98-5p	blood	Down	HMGA2	promoted PTC cell growth, inhibited apoptosis	-	Dg, Pg	Qiu. K et al. [145]
miR-296-5p	tissue	Down	PLK1	suppressed cell proliferation, inhibited cell clone formation, arrested the cell cycle in G2/M phase, and induced apoptosis	-	TT	Zhou. SL et al. [146]
miR-451a	cell culture	Down *	PSMB8	inhibited proliferation, EMT and induced apoptosis of PTC cells	-	TT	Fan. X et al. [147]
miR-630	tissue	Down	JAK2/ STAT3	suppressed migration and invasion of PTC cells	-	TT	Pan. XM et al. [148]

TT—therapeutic target, Dg-Diagnosis, Pg—Prognostic, Pr—Prevention, * The feature discovered in another study than the one cited.

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
