# Peer review of "MicroRNA in Papillary Thyroid Carcinoma: A Systematic Review from 2018 to June 2020"

_cancers, 2020, doi:10.3390/cancers12113118_

Round 1

Reviewer 1 Report

In the REVIEW “MicroRNA in Papillary Thyroid Carcinoma: A 3 Systematic review from 2018 to June 2020 “Hitu et al., give a overview of involvement of 21 microRNAs in papillary thyroid carcinoma (PTC) and their role in PTC carcinogenesis . The paper is well written.  However, the following suggestions are recommended to improve the quality of the manuscript:

Page 1 line 21: please add PTC achromous .

Page1 line 29: please use only PTC

Page 2 line 48: please spend some words to describe what microRNA are.

Page 2 line 54: please add references about role of microRNa in majority of human 54 malignancies.

Page 2 line 58: reference 4 refers to thyroid tumor but authors refer to cancer in general. Please correct accordingly.

Page 3 line 84. Results section. References list in the table are not in the references sections. Please add each references from 11 to 139 accordingly.

Page 10 line 102. Please add reference.

Page 13 line 273. Check very well the references list and correct number in the text accordingly.

Reviewer 2 Report

The manuscript by Liviu Hitu et al. summarize the information published in the last years about the involvement of  microRNAs in papillary thyroid carcinoma.

  1. Strong points of the article are:

Results and discussion are comprehensive and well written

  1. Starting from minor problems
  2. English language and style are fine/minor spell check requiredr:

“(…)more than 200 articles have been published on the  involvement  of  microRNAs  involvement  in  the  pathology  of  papillary  thyroid  cancer (…)’

  1. No consistency in spelling of miRNA in the table. “Ex-miR-423-5p” is the only one miRNA in table marked as extracellular (Ex) but some other miRNAs have blood source.
  2. Figure 2 does not add value to the article. Maybe table will be better than only 2 circle for this type of data.

III. Major problems:

Used only “microRNA” but not “miRNA” as keywords caused that some recent, important article are missed.

For example:

  1. Perdas E, Stawski R, Kaczka K, Zubrzycka M. Analysis of Let-7 Family miRNA in Plasma as Potential Predictive Biomarkers of Diagnosis for Papillary Thyroid Cancer. Diagnostics (Basel). 2020 Feb 28;10(3):130. doi: 10.3390/diagnostics10030130.

  1. Jiang K, Li G, Chen W, Song L, Wei T, Li Z, Gong R, Lei J, Shi H, Zhu J. Plasma Exosomal miR-146b-5p and miR-222-3p are Potential Biomarkers for Lymph Node Metastasis in Papillary Thyroid Carcinomas. Onco Targets Ther. 2020 Feb 13;13:1311-1319. doi: 10.2147/OTT.S231361. eCollection 2020.

Both paper relate to miRNA already included in analysis (let-7, mir-222) but in plasma and this compare (tissue/plasma) might be interesting. Please check other miRNA in this regard.

Reviewer 3 Report

Dear authors,

the review entitled "MicroRNA in Papillary Thyroid Carcinoma: A Systematic review from 2018 to June 2020" summarizes the information published in the last two and a half years about the involvement of microRNAs in papillary thyroid carcinoma and to understand the perspective offered by the new findings. This review, although it could have a potential interest for the scientific community, is difficult to read. Specifically:

  1. the periods are often too long and not linked to each other so that the meaning of the sentence is lost;
  2. often different verb forms are used (e.g. present / past in the same sentence or in 2 successive sentences;
  3. acronyms are not uniform throughout the text (e.g. microRNA and miRNA);
  4. in the table the verb forms  must all be standardized in the same form (eg inhibhits, suppression, influenced, etc ....);
  5. the conclusions must be completely rewritten.     

For these reasons, in my opinion I believe that the work cannot be accepted in its current form is and I suggest resubmission after thorough review.

Round 2

Reviewer 1 Report

Abstract

line 24: "Main microARN" it is  a mistake. Please pay attention and delete.    

line 30: use only acronym. 

Introduction:

line 55: delete PTC between references 3 and 4.   

In general, i suggested to check all manuscript because there are grammar and some other errors (line spicing)
